# Enhancing the Performance of Nanocrystalline SnO_2_ for Solar Cells through Photonic Curing Using Impedance Spectroscopy Analysis

**DOI:** 10.3390/nano14181508

**Published:** 2024-09-17

**Authors:** Moulay Ahmed Slimani, Jaime A. Benavides-Guerrero, Sylvain G. Cloutier, Ricardo Izquierdo

**Affiliations:** Département de Génie Électrique, École de Technologie Supérieure, 1100 Rue Notre-Dame Ouest, Montréal, QC H3C 1K3, Canada; moulay-ahmed.slimani.1@ens.etsmtl.ca (M.A.S.); jaime-alberto.benavides-guerrero.1@ens.etsmtl.ca (J.A.B.-G.); sylvaing.cloutier@etsmtl.ca (S.G.C.)

**Keywords:** impedance spectroscopy, photonic curing, SnO_2_, dark injection current transient, photo-Celiv

## Abstract

Wide-bandgap tin oxide (${\mathrm{SnO}}_{2}$) thin-films are frequently used as an electron-transporting layers in perovskite solar cells due to their superior thermal and environmental stabilities. However, its crystallization by conventional thermal methods typically requires high temperatures and long periods of time. These post-processing conditions severely limit the choice of substrates and reduce the large-scale manufacturing capabilities. This work describes the intense-pulsed-light-induced crystallization of ${\mathrm{SnO}}_{2}$ thin-films using only 500 μs of exposure time. The thin-films’ properties are investigated using both impedance spectroscopy and photoconductivity characteristic measurements. A Nyquist plot analysis establishes that the process parameters have a significant impact on the electronic and ionic behaviors of the ${\mathrm{SnO}}_{2}$ films. Most importantly, we demonstrate that light-induced crystallization yields improved topography and excellent electrical properties through enhanced charge transfer, improved interfacial morphology, and better ohmic contact compared to thermally annealed (TA) ${\mathrm{SnO}}_{2}$ films.

## 1. Introduction

Electron-transporting layers (ETLs) are critical components in most optoelectronic device architectures, including perovskite solar cells (PSCs). These PSC devices rely on organic–inorganic perovskite materials to efficiently absorb light and generate charge carriers [1,2,3]. ETL layers are essential for promoting efficient electron transport, block holes, align energy levels, and ultimately enhance the efficiency and stability of perovskite solar cells. Choosing appropriate ETL materials is essential for the performance of PSCs. Typical ETL materials require processing between 150 and 500 °C, resulting in higher processing times and energy costs. Most importantly, this prevents their integration on most low-cost substrates that require processing temperatures below 150 °C [4,5]. In this context, intense pulsed light annealing, also sometimes referred to as photonic curing (PC) [6], is an emerging technique that is ideally suited for large-scale manufacturing as is relies on short, high-intensity light pulses to anneal materials selectively and rapidly [7,8]. In this process, the optical energy absorbed by the active material can sustain carefully controlled light-induced annealing with minimal substrate damage. As a result, even metals with relatively high melting points can be successfully sintered on low-cost plastic- or paper-based substrates [9,10,11]. As such, this technique is also especially well-suited for roll-to-roll (R2R) manufacturing [12]. ${\mathrm{SnO}}_{2}$ metal-oxide thin-films were first utilized as ETLs for perovskite-based solar cells nearly a decade ago [13,14]. They have since emerged as the preferred material for PSCs over ${\mathrm{TiO}}_{2}$ and ZnO due to their large band gaps, higher charge mobilities, and better stabilities under ambient conditions [15,16,17]. A few years later, ${\mathrm{SnO}}_{2}$ films were photonically annealed in just 20 ms, enabling the fabrication of PSCs with reduced hysteresis and a 15% power conversion efficiency [9]. However, these previous studies did not address the effect of photonic curing on the electronic properties of SnO^2^ films. To investigate this, we used impedance spectroscopy (IS), which is a rapid technique for evaluating these properties. IS is a powerful tool to shed light on the kinetic processes taking place within electrochemical systems [18,19]. During measurement, a small alternating current (AC) signal is coupled with a direct current (DC) voltage and is applied to the device. The phase difference between the DC voltage and AC current is measured over a wide frequency range to identify the various physical effects in the device. As a result, IS measurements can assess the physical and chemical processes of various types of devices, including optoelectronic devices, fuel cells, and solid-state batteries [20]. IS is a non-destructive [14,21,22] tool that can be effectively used to optimize the stability and performance of these devices by characterizing their charge transport properties [18,23]. Typically, the IS measurements exhibit two arcs corresponding to low-frequency (*LF*) and high-frequency (*HF*) responses, respectively [24,25]. The series resistance (*Rs*), charge-transfer resistance ($R_{CT}$), and parallel capacitance can be determined from the *HF* and *LF* responses.

This work explores the impact of the photonic curing parameters on thin-film ${\mathrm{SnO}}_{2}$ properties using IS and photocurrent characteristic analysis to unveil and control the ionic and electronic kinetics within the treated ${\mathrm{SnO}}_{2}$ layer. As we demonstrate, this improved understanding and control leads to enhanced electronic properties with great potential for improved perovskite solar cell manufacturability.

## 2. Experimental Section

Commercial patterned fluorine-doped tin Oxide (FTO) substrates (SHENZHEN HUAYU UNION TECHNOLOGY, Shenzhen, China, resistance: 7 Ohm/sq) doped with fluorine are cleaned using a sequential process of 10 min each in an ultrasonic bath with DI water, acetone, and isopropyl alcohol (IPA). After drying with a nitrogen spray gun, residual organic contaminants are removed by performing a 15 min
$O_{2}$ plasma treatment (Plasma Etch, Carson City, NV, USA, PE-100LF). To prepare the ${\mathrm{SnO}}_{2}$ solution, a colloidal precursor of ${\mathrm{SnO}}_{2}$ obtained from Alfa Aesar (15% in $H_{2}O$ colloidal dispersion CN: 044592.A3) is diluted with DI water to a concentration of 3% by volume. The ${\mathrm{SnO}}_{2}$ solution is spin-coated onto the clean FTO substrate in one step in air 3000 rpm for 30 s. The edges of the FTO electrodes are then cleaned with a dry cotton swab to enable electrical and IS measurements (Figure 1). For TA, ${\mathrm{SnO}}_{2}$ films are annealed using a hot plate at 150 °C for 30 min under ambient air. For photonic curing, each sample is treated using a Novacentrix PulseForge system (500 V/3 A) power supply with 3 capacitors providing radiant energy greater than 20 J.cm^−2^ using a lamp system (7.6 cm × 60.8 cm) with an illumination area of 300 mm × 75 mm. The light source ensures uniform curing over a large area and delivers short (20 μs to 100 ms) but intense light pulses from a broadband xenon flash lamp (200–1500 nm). A Paois (Fluxim AG, SN:20121 Winterthur, Switzerland) tool is used for all electrical and IS measurements. SEM (SU8230 Hitachi) and AFM (Bruker, MultiMode8, Billerica, MA, USA ) are used for topography inspection. For impedance spectroscopy, the FTO edges that are used as electrodes are connected to the Paois to measure the impedance over a range of frequencies (10 Hz to 10 MHz) in the dark at 0.07 V perturbation at room temperature. Impedance data can be analyzed using Nyquist and Bode plots to interpret electrochemical properties such as the charge transfer resistance, capacitance, and dielectric properties. The temperature is simulated using NovaCentrix SimPulse software, this simulation package is standard on the PulseForge Version 3, Austin, TX, USA. The configuration is modeled as follows (from bottom to top): aluminum chuck, 6 mm; glass, 2.2 mm; and FTO, 600 nm. The thicknesses of the glass and FTO layers are taken from the manufacturers. X-ray diffraction (XRD) is done using a Bruker D8 Advance (Billerica, MA, USA), and optical absorbance is done using a UV–Vis–NIR spectrophotometer from Perkin Elmer (Waltham, MA, USA).

## 3. Results and Discussion

After deposition of colloidal ${\mathrm{SnO}}_{2}$ films using the protocol, samples are post-processed using varying pulse durations and energy densities using the methodology described in the Section 2. To investigate the impact of PC on the electrical properties of ${\mathrm{SnO}}_{2}$ films, we conduct flash annealing for pulse durations of 500, 1500, 2500, and 3500 μs, followed by photocurrent measurements. This allows us to optimize our photonic annealing parameters and define the high photoconductivity range for ${\mathrm{SnO}}_{2}$ films. Photocurrent analysis is used to map the different zones’ photoconductivity. Pulses ranging from 500 to 3500 μs are utilized to complete the photo-responsivity characterization. Figure 2a shows the I–V responses in the dark and under illumination for two samples photonically treated using a pulse duration of 2500 μs and, respectively, 2 J.cm^−2^ and 4 J.cm^−2^. A low photo-responsivity indicates that the illumination and dark curves approach the overlap limit, while a high photo-responsivity indicates a clear offset (more than 0.5 order of magnitude) between the I–V characteristics in the dark and under illumination. Based on such measurements, Figure 2b displays a photo-responsivity map for samples photonically treated using different pulse durations vs. energy densities. To shed light on these results, IS and SEM characterizations are conducted. ${\mathrm{SnO}}_{2}$ is highly transparent, which makes photonic curing difficult [26]. To mitigate this problem, we use substrates with FTO patterns that act as a structural support and a stable base for the growth of ${\mathrm{SnO}}_{2}$ nanoparticles. This helps promote the transmission of the heat generated when light is absorbed by the nanoparticles [27], which can increase the local temperature around the nanoparticles and promote the recrystallization process. FTO substrates exhibit rougher surfaces than glass [28], promoting superior adhesion and growth of ${\mathrm{SnO}}_{2}$ nanoparticles [29]. Their conductivity enhances the electrical properties of the resulting ${\mathrm{SnO}}_{2}$ films. The FTO substrate’s roughness directly influences both the diameter and alignment of the ${\mathrm{SnO}}_{2}$ nanoparticles [30]. Areas with FTO patterns acting as a blanket allow for changes in nanoparticle recrystallization depending on the energy density used.

Figure 2c displays SEM images of the bare FTO substrate, and Figure 2d–f show ${\mathrm{SnO}}_{2}$ films deposited on FTO and photonically treated using energy densities of 0.15, 2.06, and 2.46 J.cm^−2^, respectively. As the energy density is increased from 0.15 to 2.06 to 2.46 J.cm^−2^ while using 1500 μs pulse durations, the ${\mathrm{SnO}}_{2}$-covered films appear increasingly granular, while the distinct grain boundaries that were clearly observed in the FTO/glass film are less apparent. The recrystallization of the ${\mathrm{SnO}}_{2}$ film follows the substrate topography well, revealing the underlying FTO grain profile. This process indicates that higher energy densities lead to improved film–substrate adhesion and more pronounced exposure of the underlying grain structure. The photonic curing of ${\mathrm{SnO}}_{2}$ wet films enables water evaporation and subsequent crystallization of ${\mathrm{SnO}}_{2}$ nanoparticles [31]. The degree of crystallization greatly affects the photoconductivity of ${\mathrm{SnO}}_{2}$ films and their ability to carry charge carriers [32,33]. A film’s properties largely depend on two independent parameters: the energy density and the pulse duration of the pulsed light.

To obtain quantitative information and to better understand the surface morphology and roughness, we also conduct AFM analyses on samples subjected to different types of annealing treatments. Figure 2g shows the surface roughness of the film samples for a scan area of 5 × 5 $\mu m^{2}$. It highlights the improvement in surface topography after optimal photonic treatment with ${\mathrm{SnO}}_{2}$, with a root-mean-square roughness of 14.01 nm, compared to 45.57 nm for the thermally annealed sample. Roughness is defined as the microscopic and macroscopic variations on a material’s surface [34]. It measures the irregularities present on the surface. These variations can have a significant impact on the physical and chemical properties of materials, such as the recombination rates in ETL films for solar cells. Low-roughness films can reduce the recombination rate and thus improve performance [35]. The morphological effect of PC processing can be beneficial in terms of device performance. This significant improvement underscores another important advantage of photonic treatment for enhancing the quality of ${\mathrm{SnO}}_{2}$ as an electron transport layer (ETL) in perovskite solar cells.

This section focuses on the variation of IS results for ${\mathrm{SnO}}_{2}$ films treated with different energy densities and pulse durations of 500, 1500, 2500, and 3500 μs. For these measurements, the ${\mathrm{SnO}}_{2}$ film is deposited onto FTO glass, and its electrochemical behavior can be represented by an equivalent circuit that produces a semicircle on the Nyquist diagram. Figure 3a–d displays IS results for ${\mathrm{SnO}}_{2}$ samples treated using these different pulse durations and energy densities. When the pulse duration is fixed and the energy density is increased, the semicircle decreases until it reaches its minimum, and then the arc widens. The frequency response exhibits two distinct behaviors. At high frequencies (*HF*), it is dominated by the resistance attributed to electronic transport ($R_{CT}$). At low frequencies (LF), it is dominated by the recombination resistance ($R_{rec}$) related to ionic diffusion and charge accumulation at the contacts [36,37]. In Figure 3, it corresponds to the second semicircle inclined at 45° to the real axis in the Nyquist graph [38]. The semi-circle in the high-frequency region is generally related to the counter-electrode and its interface [39]. A smaller half-circle suggests a lower $R_{CT}$ and better photoconductivity of the device. These Nyquist plots suggest that our devices’ equivalent circuits can be accurately modeled by a resistor–capacitor (RC) pair in the dark AC regime [40]. As such, the interface contribution can be derived from the equivalent circuit’s parameters [41]. The series resistance (Rs) can be obtained by measuring the shift of the semi-circle from the origin along the horizontal axis [42]. However, the time constant related to the physical phenomena dominating at both the low and high frequencies is described by $\tau_{HF}.\omega_{HF}=1$ and $\tau_{LF}.\omega_{LF}=1$, with $\omega_{HF,LF}=2\pi \;\cdot \;f_{max,HF,LF}$ [40]. The time constants can be deduced from the IS results by identifying the peak of the semicircle, which corresponds to the maximum frequency, or by calculating τ=Req.Ceq, as shown in Table 1.

Figure 4a compares the Cole–Cole plots for films that are photonically treated using 500, 1500, 2500, and 3500 μs pulses with respective energy densities of 0.52, 2.45, 3.44, and 3.55 J.cm^−2^ with a typical film sample crystallized using standard thermal annealing.Clearly, the physical and chemical properties of the resulting ${\mathrm{SnO}}_{2}$ films appear greatly affected by the pulse duration and energy density. When the pulse duration is 3500 μs and the energy density is 3.55 J.cm^−2^, the high-frequency arc is smallest, suggesting that the film is less resistive and facilitating charge transfer. In comparison, the thermally annealed sample exhibits a larger semicircle than all of the photonically treated samples. This suggests increased imaginary impedance associated with a decrease in charge transfer. Figure 4b–d compare the imaginary impedance, capacitance, and conductance versus the frequency for the best thermally annealed and the best photonically treated films for the conditions 3.55 J.cm^−2^ and 3500 μs. In Figure 4b, the high-frequency (*HF*) peaks appear between $10^{5}$–$10^{6}$ Hz for both samples. The response time can be obtained by taking the inverse of the peak frequency from the imaginary impedance graph. Table 1 presents the IS parameters extracted from the spectra. There, the $R_{CT}$ value for the thermally annealed sample is roughly twice the value achieved using optimal photonic curing conditions. This suggests that the ${\mathrm{SnO}}_{2}$/FTO interface provides a low $R_{CT}$ under the effect of photonic annealing, which facilitates charge carrier transport. The resulting time constant is 0.8 μs for the thermally annealed film, compared to 0.38 μs for the optimal photonic curing conditions. This suggest that photonically induced crystallization promotes a faster response time, resulting in low recombination and more dominant ionic diffusion behavior [43,44]. At low frequencies, the thermally annealed device does not exhibit any measurable peak, which is consistent with the presence of the single semicircle in Figure 4b. In contrast, the impedance plot of the photonically treated device is curved at low frequencies, explaining the start of the second semicircle in this region. Frequency, time constant, and conductivity values are good indicators of process kinetics [45,46]. Indeed, the dark IS can be directly related to the carrier density, mobility, and conductivity [38]. The temperature simulation results using the photonic annealing parameters shown in Figure 4e reveal a relationship between the energy density, pulse duration, and resulting temperature of the ${\mathrm{SnO}}_{2}$ film. As the energy density increases from 0.52 to 3.55 J.cm^−2^, the temperature increases from 122 to 364 °C then decreases to 329 °C for the film treated with an energy density of 3.55 J.cm^−2^ and a pulse duration of 3500 μs. These parameters are crucial to determine the energy transferred to the ${\mathrm{SnO}}_{2}$ film, but they show a non-linear trend with temperature. Figure 4f shows X-ray diffraction (XRD) measurements of the thermally and photonically annealed ${\mathrm{SnO}}_{2}$ films. The prominent peaks are determined to correspond to (110), (101), (200), (211), (220), and (002), confirming the tetragonal crystal structure of ${\mathrm{SnO}}_{2}$ for both the TA and PC films [47,48,49].

Figure 4c,d show capacitance and conductivity evolutions as a function of the operation frequency. Figure 4c illustrates two distinct capacitance behaviors, each corresponding to a specific polarization process. This distinction makes it possible to identify specific capacitive processes directly from the plot [50,51]. The high-frequency capacitance $C_{HF}$ (above 100 kHz) exhibits a plateau in the order of 1 pF for both thermally and photonically treated devices and is rather similar for both annealing processes. This region represents the geometric capacitance and is due to the intrinsic dielectric polarization of the ${\mathrm{SnO}}_{2}$ layer [50]. However, photonic treatment achieves higher capacitance values at low frequencies (below 1 KHz) compared to the thermally annealed device. This is primarily due to the accumulation of charges or ions [52,53] resulting from the polarization of the interfaces between the ${\mathrm{SnO}}_{2}$ layer and the electrodes. At low frequencies, the increase in capacitance is dominated by ionic movement in the dark and electronic movement in the light [54,55]. In circuits that exhibit capacitive behavior, the capacitor offers less resistance to the flow of alternating current as the frequency increases. Accordingly, Figure 4d shows increases in conductance for both devices in the high-frequency region. This behavior is consistent with that of semiconductors, where capacitance and conductance vary inversely [56,57,58].

The optical properties of the prepared samples are characterized by UV–Vis absorption spectra. As shown in Figure 5a, the transmittance of PC-treated films is higher than that of TA-treated films, which is desirable for solar cell applications. ${\mathrm{SnO}}_{2}$ is a direct bandgap (BG) semiconductor; its BG can be calculated using a Tauc plot [59], as shown in Figure 5b. The calculated BGs are 3.45 eV and 3.43 eV for the TA- and PC-treated films, respectively, which explains why the TA film is slightly more transparent than the PC film. Measurements in Figure 5c,d compare the dark injection transients for the photocurrent rise and decay for the thermally and photonically treated (3.55 J.cm^−2^, 3500 μs) samples. This time-of-flight technique is useful for determining majority carrier mobility and trapping, especially in thin-films [60]. Figure 5c illustrates that the current for the photonically treated film rises to 2.7 mA, compared to 2.3 mA for the thermally annealed film. The current also increases more rapidly in the photonically treated sample, reflecting the interrelationship between charge carrier generation and recombination. Therefore, the rapid increase in current for the PC sample can be attributed to the fast accumulation of photogenerated carriers [61]. Figure 5d compares the decay of the transient current. After reaching its maximum, the current decay depends on the charge capture coefficient [62]. The decay graph illustrates the speed of charge recombination after being excited by a 1.2 V pulse voltage. A shorter carrier lifetime suggests faster recombination and a high carrier capture rate, which implies more rapid current decay for the thermally annealed sample. In contrast, photonic curing yields a lower recombination rate, resulting in slower decay and longer current holding times. The photogeneration and recombination processes have a significant impact on the density and mobility of charge carriers. Figure 5e compares the charge mobility using the photo-CELIV technique using the following expression [63,64,65]:(1)$$\mu =\frac{2d^{2}}{3A.t_{max}^{2}\left(1+0.36\frac{\Delta }{J_{max}}\right)}$$
where *d* is the ${\mathrm{SnO}}_{2}$ film thickness, *A* is the slope of the extraction voltage ramp, $t_{max}$ is the time related to the current peak, and Δ is the difference between the maximum current and the displacement current plateau. Photo-CELIV is a technique used to extract the charge mobility by illuminating the device. The measurement displays the current overshoot and the time at which the current reaches its maximum, which is an essential parameter for quantifying mobility. However, it should be noted that Photo-CELIV only measures fast carriers and cannot distinguish between the mobility of electrons and holes. The Photo-CELIV measurements for the film after optimized photonic treatment yield $4.56\times 10^{-2}\;V\;{cm}^{2}s^{-1}$, compared with $3.66\times 10^{-2}\;V\;{cm}^{2}s^{-1}$ for the thermally annealed film. This measurement does not precisely reflect the mobility of the ${\mathrm{SnO}}_{2}$ material. However, it serves as a characterization for comparing the fastest or maximum carrier mobility values. This higher maximum mobility compared to thermal annealing is consistent with previous results.

## 4. Conclusions

In summary, we propose an optimized photonic annealing approach to improve the electrical properties of ${\mathrm{SnO}}_{2}$ thin-films compared to standard annealing. ${\mathrm{SnO}}_{2}$ thin-films play an essential role in emerging device architectures, especially as the electron-transporting layer (ETL) for perovskite-based solar cells. We use impedance spectroscopy to analyze the electrical behavior of ${\mathrm{SnO}}_{2}$ films in the dark. The results indicate that the impedance spectroscopy response depends significantly on both the energy density and the pulse duration and shed light on the resulting ionic and electronic transfer. Additionally, we demonstrate that photonic treatment yields ${\mathrm{SnO}}_{2}$ layers with enhanced electrical performance and a significantly reduced manufacturing time compared to standard thermal annealing. This would be a great advantage for large-scale manufacturing of better and cheaper perovskite-based solar cells.

## Figures and Tables

**Figure 1 nanomaterials-14-01508-f001:**
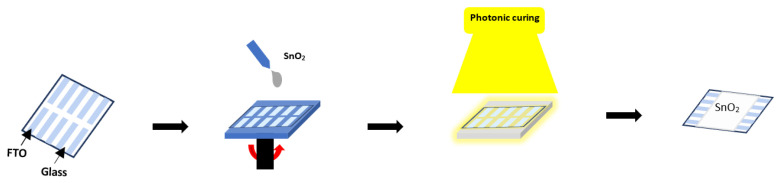
Illustration of the ${\mathrm{SnO}}_{2}$ sample fabrication process.

**Figure 2 nanomaterials-14-01508-f002:**
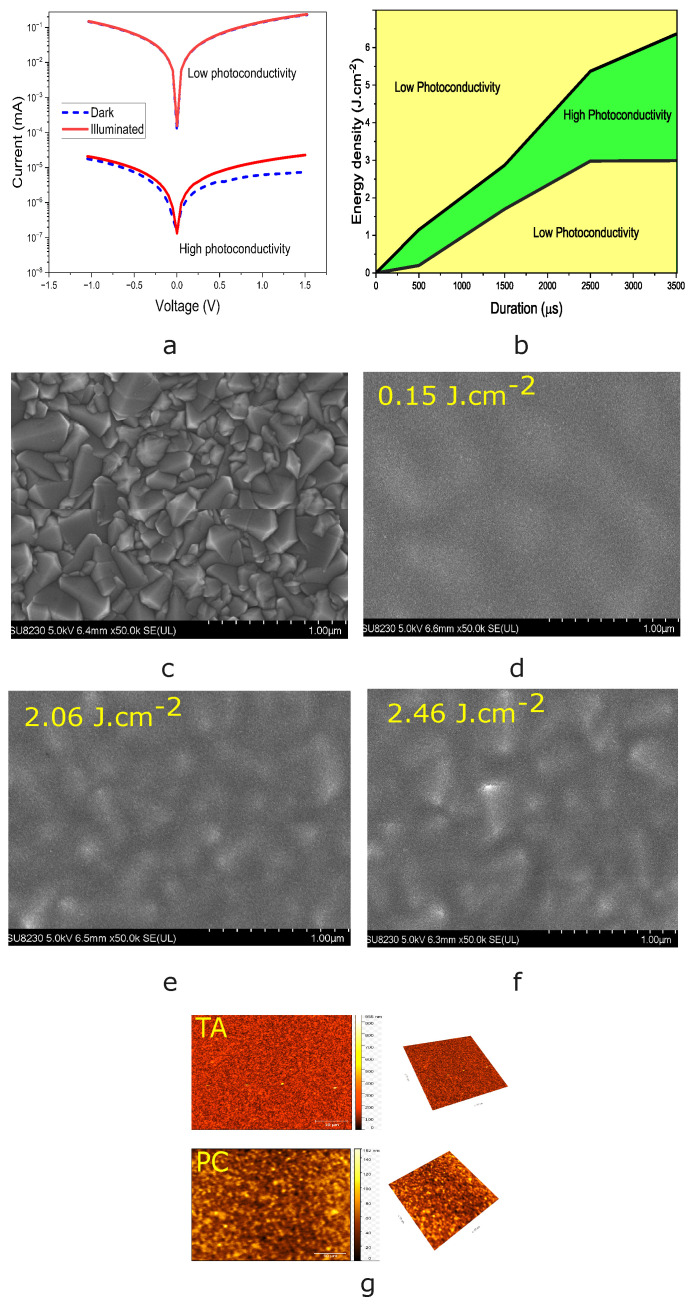
(**a**) I−V responses in the dark and under illumination for two samples photonically treated using a pulse duration of 2500 μs and, respectively, 2 and 4 J.cm^−2^. (**b**) Photo-response map for samples photonically treated using different pulse durations vs. energy densities based on the criterion in Figure 2a. (**c**) SEM images of FTO/glass. (**d**–**f**) SEM images of PC of ${\mathrm{SnO}}_{2}$ samples on FTO/glass. (**g**) Atomic force microscopy (AFM) images in 2D and 3D of thermally and photonically annealed samples.

**Figure 3 nanomaterials-14-01508-f003:**
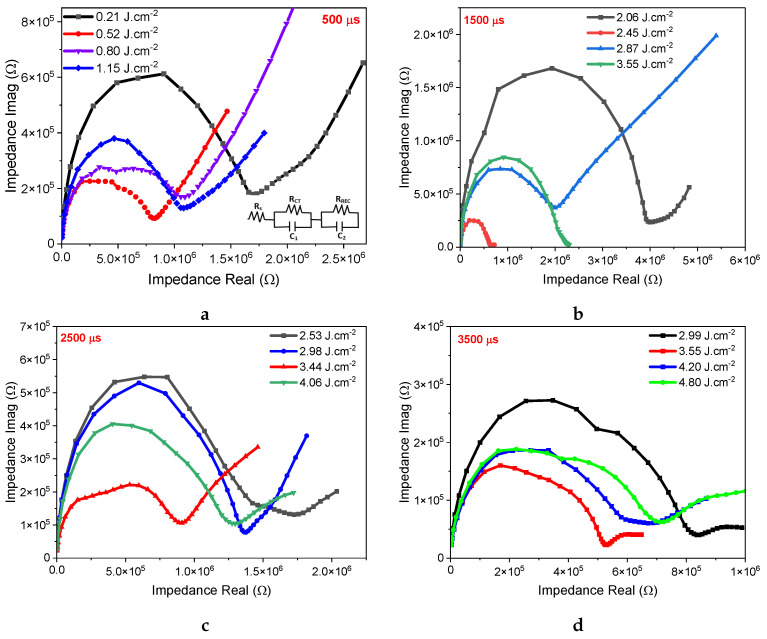
Imaginary versus real components of impedance for photonically annealed films with pulse durations of (**a**) 500, (**b**) 1500, (**c**) 2500, and (**d**) 3500 μs, respectively.

**Figure 4 nanomaterials-14-01508-f004:**
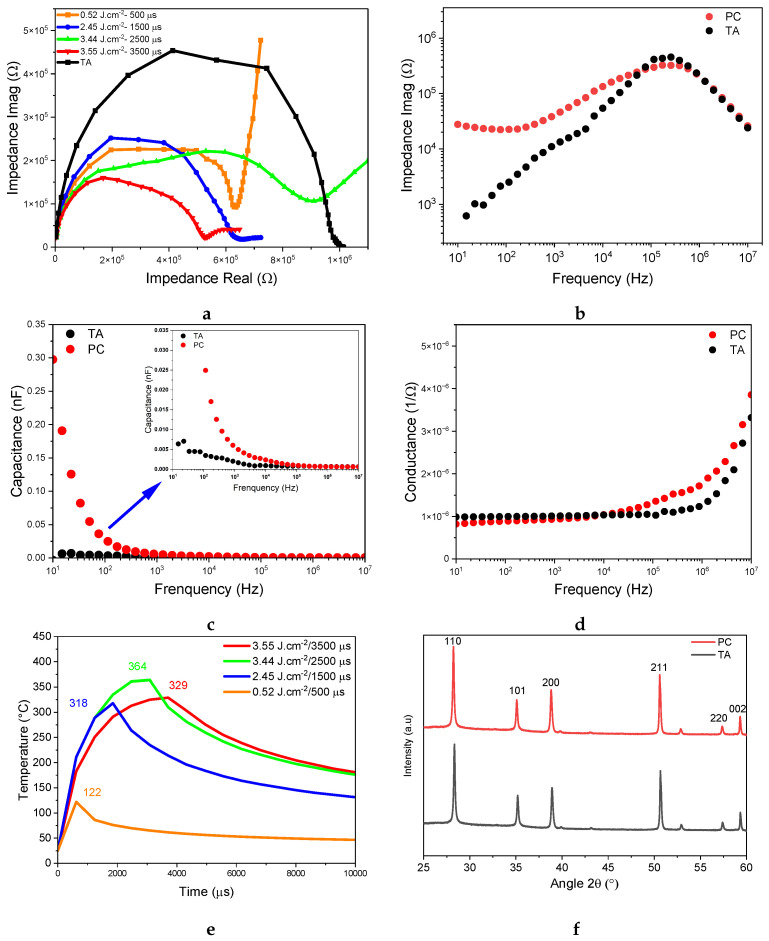
(**a**) Cole–Cole plot for films thermally and photonically treated using 500, 1500, 2500, and 3500 μs with energy densities of 0.52, 2.45, 3.44, and 3.55 J.cm^−2^, respectively. (**b**–**d**) Comparison of imaginary impedance, capacitance, and conductance vs. frequency for typical thermally annealed and photonically treated samples. (**e**) SimPulse simulations of temperature profiles of photonically annealed SnO^2^ film Cole–Cole plots for films photonically treated using 500, 1500, 2500, and 3500 μs with energy densities of 0.52, 2.45, 3.44, and 3.55 J.cm^−2^, respectively. (**f**) XRD spectra of thermally and photonically annealed ${\mathrm{SnO}}_{2}$ films at 3.55 J.cm^−2^
μs.

**Figure 5 nanomaterials-14-01508-f005:**
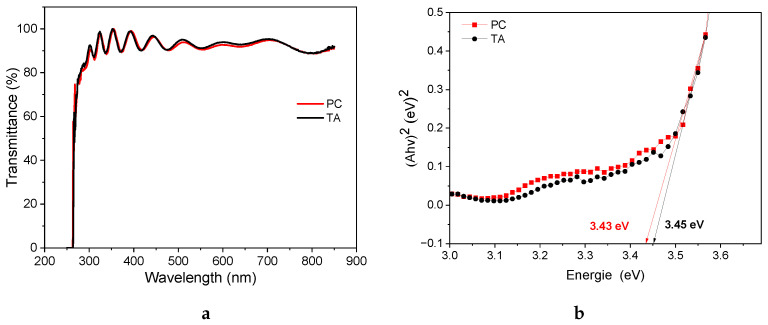
(**a**,**b**) Transmittance spectra and Tauc plots of thermally and photonically annealed ${\mathrm{SnO}}_{2}$ samples. (**c**,**d**) Dark injection transients for the photocurrent rise and decay for the thermally and photonically treated samples. (**e**) Charge mobility using the photo-CELIV technique for the thermally and photonically treated samples.

**Table 1 nanomaterials-14-01508-t001:** IS parameters extracted from the Nyquist plots for thermally annealed and photonically treated samples at 0 V in dark conditions with 0.07 V perturbations. Photonic treatment is performed using a 3500 μs pulse duration at 3.55 J.cm^−2^ energy density.

Device	Rs (kΩ)	$R_{\mathrm{CT}}$ (MΩ)	Ceq (pF)	$\tau_{\mathrm{HF}}$ (μs)
Thermally annealed	1.96	0.99	0.88	0.87
Photonically treated	3.06	0.49	0.78	0.38

## Data Availability

The data that support the findings of this study are available from the corresponding author upon reasonable request.

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
