# Peer review of "Enhancing the Performance of Nanocrystalline SnO2 for Solar Cells through Photonic Curing Using Impedance Spectroscopy Analysis"

_nanomaterials, 2024, doi:10.3390/nano14181508_

Round 1

Reviewer 1 Report

Comments and Suggestions for Authors

          The paper presents original results on an approach to crystallization of SnO2 layers using an intense pulsed light with the pulse duration below 1 s. This method has an advantage over a prolonged thermal annealing of the SnO2 material used otherwise for device applications such as photovoltaic cells.  A a colloidal precursor of SnO2 was used as a starting material.

           As a reviewer I would recommend

  1. Article title. It might be useful to introduce some words showing that the performance of the SnO2 is related to solar cells.
  2. Last keyword: seems to be a mistype
  3. Experimental section, line 67 states «For TA, SnO2 films were annealed». The “TA” term was not introduced earlier in the manuscript.
  4. Experimental section, line 68-69. Irradiation source is not described enough. Peak wavelength and spectral distribution are of great importance.
  5. Line 105. Upon discussion results in Fig 2 the authors state  “helps in binding the films to the substrate”. While they have not carried out any measurements in order to characterize the strength of the binding.
  6. Line 115, upon discussion on Figure 2f. The authors make a controversial conclusion. The film uniformity is not improved in case of photonic treated film compared with thermal (TA) treated. The photonic treated films demonstrate larger grains, on the opposite.
  7. The results on film characterization by XRD (structural) or by XPS, for instance (atomic composition) should be provided in order to make reliable conclusions on the properties of such newly prepared material. With no such characterization the manuscript quality can not be considered as better than average.

  I therefore can recommend this manuscript for publication only after major amendments.

Regards, Reviewer

---------

Author Response

Reviewer 1

We would like to thank the reviewer for his in-depth criticism and comments, which have improved this manuscript and made it more relevant. All modifications to the original manuscript are highlighted in yellow for new inclusions. Where necessary, the original line number of the manuscript is indicated to make it easier to locate the modified text in the original submission. A version with highlighted changes is included in the unpublished material and a final version of the manuscript is uploaded with this response.

1. Article title. It might be useful to introduce some words showing that the performance of the SnO2 is related to solar cells.

The title was changed to “Enhancing the Performance of Nanocrystalline SnO2 for Solar Cells through Photonic Curing Using Impedance Spectroscopy Analysis’’

2. Last keyword: seems to be a mistype

We change the word “spectroscopie’’ to spectroscopy (line 13)

3. Experimental section, line 67 states «For TA, SnO2 films were annealed». The “TA” term was not introduced earlier in the manuscript.

The word TA is introduced in the “Abstract section’’, line 12.

4. Experimental section, line 68-69. Irradiation source is not described enough. Peak wavelength and spectral distribution are of great importance.

We have expanded this part by providing more details about the lamp spectrum. The changes are shown in lines (68-73).

5. Line 105. Upon discussion results in Fig 2 the authors state “helps in binding the films to the substrate”. While they have not carried out any measurements in order to characterize the strength of the binding.

We've reworded this section to make it clearer and more concise lines (116-119)

6. Line 115, upon discussion on Figure 2f. The authors make a controversial conclusion. The film uniformity is not improved in case of photonic treated film compared with thermal (TA) treated. The photonic treated films demonstrate larger grains, on the opposite.

We thank the reviewer for pointing out this omission from our part. We have reworded this section to make it clearer and more unambiguous. In this section, we did not address the difference in grain size for the two treatments but focused on the effect of the annealing process on the roughness of SnO2 films. Low roughness films can reduce recombination rates and therefore improve performance when incorporated into solar cell configurations, for example. The morphological effect of PC processing could be beneficial in terms of device performance. The change with references is shown on lines (128-135)

7. The results on film characterization by XRD (structural) or by XPS, for instance (atomic composition) should be provided in order to make reliable conclusions on the properties of such newly prepared material. With no such characterization the manuscript quality can not be considered as better than average.

We thank the reviewer for bringing this to our attention. XRD characterization and analysis are added (figure 4f) and lines (195-198)                        

Reviewer 2 Report

Comments and Suggestions for Authors

General comment:

In this manuscript, the authors presented a study of the photonic treatment yields SnO2 layers with enhanced electrical performance, which can provide a great advantage for large-scale manufacturing of better and cheaper perovskite-based solar cells. This study was interesting. Accordingly, I would like to recommend this article after the comments below addressed.

Comment 1:

The authors should provide the first use of an abbreviation immediately before or after the expanded form.  For instance, TA….

Comment 2:

What is the real temperature on FTO right after the flash annealing?

Comment 3:

The authors should mention how to use impedance spectroscopy to measure SnO2 film or solar cell (?) in detail.

Comment 4:

The authors should provide the references for equivalent circuits to fit impedance spectroscopy. The references related to charge mobility using the photo-CELIV technique should be also given.

Author Response

Reviewer 2

 We would like to thank the reviewer for his in-depth criticism and comments, which have improved this manuscript and made it more relevant. All modifications to the original manuscript are highlighted in yellow for new inclusions. Where necessary, the original line number of the manuscript is indicated to make it easier to locate the modified text in the original submission. A version with highlighted changes is included in the unpublished material and a final version of the manuscript is uploaded with this response.

Comment 1:

The authors should provide the first use of an abbreviation immediately before or after the expanded form.  For instance, TA….

 TA is introduced in the “Abstract section’’, line 12.

Comment 2:

What is the real temperature on FTO right after the flash annealing?

We thank the reviewer for bringing this to our attention. We have added the real temperatures for each photonic annealing treatment (figure 4e). The measurement was detailed in the experimental section (79-82)

Comment 3:

The authors should mention how to use impedance spectroscopy to measure SnO2 film or solar cell (?) in detail.

We thank the reviewer for bringing this to our attention. Many details are provided in the experimental section. Lines (75-79)

Comment 4:

The authors should provide the references for equivalent circuits to fit impedance spectroscopy. The references related to charge mobility using the photo-CELIV technique should be also given.

We thank the reviewer for pointing out this omission.

  • We added references for equivalent circuit to fit impedance spectroscopy, line 50
  • We added references related to charge mobility using the photo-CELIV technique, line 240. Other references have been added in the modified sections.

Reviewer 3 Report

Comments and Suggestions for Authors

The manuscript 'Enhancing Performance of Nanocrystalline SnO2 by Photonic Curing Using Impedance Spectroscopy Analysis' is well constructed and useful for energy applications. 

The authors provided different characterization techniques to justify their findings. However, the authors should provide the following information before accepting the manuscript.

1. XRD of the samples to understand the crystalline property compared to thermal annealing.

2. It will be interesting to see the effect of photonic curing by UV-Vis spectroscopy compared to thermal annealing. 

3. Also, I suggest the authors provide band gap data of the samples.

Author Response

Reviewer 3

We would like to thank the reviewer for his in-depth criticism and comments, which have improved this manuscript and made it more relevant. All modifications to the original manuscript are highlighted in yellow for new inclusions. Where necessary, the original line number of the manuscript is indicated to make it easier to locate the modified text in the original submission. A version with highlighted changes is included in the unpublished material and a final version of the manuscript is uploaded with this response.

  1. XRD of the samples to understand the crystalline property compared to thermal annealing.

We thank the reviewer for bringing this to our attention. XRD characterization and analysis are added (figure 4f) and lines (195-198)                        

  1. It will be interesting to see the effect of photonic curing by UV-Vis spectroscopy compared to thermal annealing. 

We thank the reviewer for pointing out this omission.

Transmittance is added (figure 5a) and analysis of this measure lines (218-220).

  1. Also, I suggest the authors provide band gap data of the samples.

We thank the evaluator for mentioning that.

We have added the BG plot (figure 5b) and the analysis of this measure lines (220-223).

Round 2

Reviewer 1 Report

Comments and Suggestions for Authors

All necessary corrections to the previous versions were made. The paper can well be accepted. 

Reviewer 3 Report

Comments and Suggestions for Authors

The authors have modified the manuscript suitably, and it can be accepted in its current state.